# To Understand Indicators of Robots' Vision Capabilities

### Hong Wang
RARE Lab
Department of Computer Science and Engineering
University of South Florida
Tampa, Florida, USA
hongw@usf.edu

### Tam Do
RARE Lab
Department of Computer Science and Engineering
University of South Florida
Tampa, Florida, USA
tamdo@usf.edu

### Zhao Han
RARE Lab
Department of Computer Science and Engineering
University of South Florida
Tampa, Florida, USA
zhaohan@usf.edu

## ABSTRACT

Study [10] indicates that humans can mistakenly assume that robots and humans have the same field of view (FoV), possessing an inaccurate mental model of a robot. This misperception is problematic during collaborative HRI tasks where robots might be asked to complete impossible tasks about out-of-view objects. To help align humans' mental models of robots' vision capabilities, we explore the design space by proposing nine FoV indicator designs in augmented reality (AR) and present an experiment design to evaluate them subjectively and objectively. Specifically, we designed these indicators from a spectrum from the head to the task space. Regarding the experiment, we propose an assembly task to assess them in terms of accuracy, task efficiency, and subjective experience like confidence and likability. Our future results will inform robot designers to better indicate robots' FoV to align humans' mental models of robots' vision capabilities.

## CCS CONCEPTS

• **Computer systems organization** → **Robotics**; • **Human-centered computing** → **Mixed / augmented reality**.

## KEYWORDS

System transparency, robot design, vision capability, field of view, augmented reality (AR), human-robot interaction (HRI)

**ACM Reference Format:**
Hong Wang, Tam Do, and Zhao Han. 2024. To Understand Indicators of Robots' Vision Capabilities. In *Proceedings of the 7th International Workshop on Virtual, Augmented, and Mixed-Reality for Human-Robot Interactions (VAM-HRI '24), March 11, 2024, Boulder, CO, USA.* 5 pages.

## 1 INTRODUCTION

Mental models are structured knowledge systems that enable people to engage with their surroundings [33]. They can influence how people perceive problems and decision-making [13], and present how individuals interact within complex systems, such as technological or natural environments [18]. In a team environment, a shared mental model improves team performance when team members have a mutual understanding of the task [14]. This is also true in human-agent teams [29], applicable to physically embodied agents like robots. Indeed, Mathieu et al. [19] found that both team- and task-based mental models were positively related to efficient team process and performance. This highlights the importance of shared mental models in shaping effective teamwork. To leverage the shared mental models, Hadfield-Menell et al. [8] proposed a cooperative inverse reinforcement learning formulation to ensure that agents' behaviors are aligned with humans' goals. Nikolaidis et al. [22] also developed a game-theoretic model of human adaptation in human-robot collaboration. These studies show that shared mental models are crucial for both human teams and human-robot teams: They enhance coordination, improve performance, and help better understand collaborative tasks.

However, in human-robot teaming and collaboration scenarios, because robots more or less resemble humans, humans can form an inaccurate mental model of robots' capabilities, leading to mental model misalignment. Frijns et al. [7] noticed this problem and proposed an asymmetric interaction model: Unlike symmetric interaction models where roles and capabilities are mirrored between humans and robots, asymmetric interaction models emphasize the distinct strengths and limitations of humans and robots.

One mental model misalignment case, related to a robot's vision limitation, is the assumption that robots possess the same field of view (FoV) as humans. Although humans have over 180° FoV, a robot's camera typically has less than 60° horizontal FoV (e.g., Pepper's 54.4° [1] and Fetch's 54° [27, 34]). This discrepancy and assumption are problematic. Specifically, Han et al. [10] investigated how a robot can convey its incapability of handing a cup that is both out of reach and out of view. Yet, participants assumed the cup was within the robot's FoV, and expected the robot to successfully hand it to them. In those cases, robots benefit from a more accurate mental model, leading to fewer explanations and clearer instructions, e.g., "the cup on the right" rather than "the cup".

In this paper, we aim to address the FoV discrepancy by exploring the design space and detailing an evaluation plan of our FoV indicator designs. Specifically, we propose nine situated augmented reality (AR) indicators (Figure 1 except for the baseline) to align human expectations with the real vision capability of robots. We are interested in AR as the robot's hardware is hard to modify, and AR allows fast prototyping to narrow down the design space. Moreover, the situatedness allows placing indicators near the eyes–which have the FoV property–and the task objects–which are negatively affected by FoV misperception–rather than, e.g., on a robot's FoV-irrelevant chest screen.

With those indicators already designed, we plan to register them onto a real Pepper robot and propose a user study to narrow down

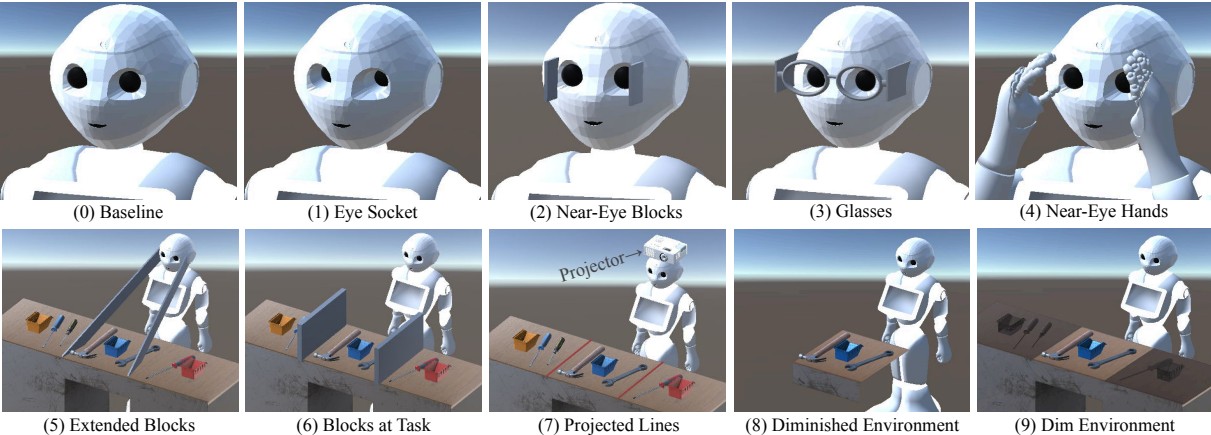

(0) Baseline    (1) Eye Socket    (2) Near-Eye Blocks    (3) Glasses    (4) Near-Eye Hands

(5) Extended Blocks    (6) Blocks at Task    (7) Projected Lines    (8) Diminished Environment    (9) Dim Environment

**Figure 1: Nine indicator designs with a baseline in the beginning, to be registered to the real robot and evaluated in a human-subjects study. The tools rendered will be replaced with our task tools later (see Figure 4). The designs are detailed in Section 4.**

and investigate the effects of those designs subjectively and objectively. Participants will assemble an airplane with a robot's help in delivering the tools that they may forget. We will record the help requests and task completion time to measure effectiveness and efficiency and ask participants to complete surveys to capture their subjective experiences. This work will help robot designers to better indicate robots' FoV to align humans' mental models of robots' vision capabilities.

## 2 RELATED WORK

### 2.1 AR for Robotics

Robotics researchers have integrated AR in multiple domains. Examples include AR systems for fault visualization in industrial robots [2], integration in robotic surgical tools [4], and AR-enhanced robotics education for interaction [25]. These studies showed the potential of AR in enhancing human-robot interactions. Particularly, Avalle et al. [2] developed an AR system to support fault visualization in industrial robotic tasks by visualizing robot's operational status and faults with icons in AR. In the study by Jost et al. [15], AR is integrated into a heterogeneous fleet management system with AR-equipped safety vests for human workers, allowing them to see where the robots are even out of sight, e.g., blocked by racks, thereby increasing the feeling of safety in warehouse environments. Finally, Das and Vyas [4] explored the integration of AR/VR with robotic surgical tools, highlighting the increased precision and user comprehension that situated AR overlays offer in complex surgeries.

### 2.2 AR Design Elements in Robotics

There are extended works using AR to make design presentations. Fang et al. [6] designed a novel AR-based interface to visualize interactive robot path planning for manipulation tasks like robot pick-and-place and path-following. Peng et al. [24] introduced an interactive system combining AR and a robotic 3D printer for real-time, hands-on modeling and fabrication, thus enhancing the design process. Walker et al. [31] discussed AR as a medium to mediate

human-robot interactions for aerial robots. The research proposed both explicit and implicit AR-based designs to visually signal robot motion intent. Results found that AR designs significantly enhanced task efficiency compared to just using physical orientation cues.

Recently, Walker et al. [32] proposed a novel taxonomy for identifying and categorizing VAM-HRI Virtual Design Elements (VDEs) from 175 research papers into four categories: Virtual Entities, Virtual Alterations, Robot Status Visualizations, and Robot Comprehension Visualizations. As detailed in Section 4, our designs fall under "Virtual Alterations – Morphological" (Figure 1.1 to 1.3) and "Virtual Entities - Environmental" (Figure 1.5 to 1.9).

### 2.3 AR for Robot Comprehension

There has been a variety of work on specifically helping users to understand robot intentions and behaviors using AR. Bolano et al. [3] investigated transparent robot behavior in close-proximity human-robot interaction, highlighting the development and implementation of an AR system that projects representations like motion plan, task state, and potential collision points. Furthermore, Rotsidis et al. [28] discussed the development of an AR software tool to debug and show the navigation goals that enhance the transparency of mobile robots. Dyck et al. [5] also showed the impact of AR visualizations on users' understanding and acceptance of robots during navigation tasks. For drones, Szafir et al. [30] explored the design space of visually communicating the directional intentions of drones using AR.

Another line of work focuses on conveying what a robot perceives about the environment to people, e.g., adding external sensor purviews [32]. For example, Kobayashi et al. [17] uses AR to overlay a humanoid robot's sensory perceptions like obstacle representations and decision-making processes of navigation onto the physical environment. The most relevant work is Hedayati et al. [12]'s. They proposed a framework for structuring the design of AR interfaces for HRI. Under this framework, they developed three teleoperation models to provide visual feedback on robot camera capabilities like real-time visual overlays, interactive interface elements, and

| Eye Space | Head Space | Transition Space | Task Space |
|---|---|---|---|
| (1), (2) | (3), (4) | (5) | (6), (7), (8) (9) |
| *Robot* | | | *Environment* |

**Figure 2: Our design spectrum based on proximity to the robot (eye to head) and the task environment.**

enhanced camera feeds. However, their work has focused on non-collocated teleoperation. For more work, please refer to Walker et al. [32]. Our work, while aligning with robot comprehension visualizations in environments, specifically aims to convey robots' vision capabilities through AR indicators.

## 3 DESIGN TAXONOMY AND SPECTRUM

As robots are physically situated in our physical world, we grouped our designs into four connected areas based on the proximity to the robot and its task environment. It formed a spectrum as shown in Figure 2. *Eye space* designs focus on the modifications at the robot's eyes, which possess the property of FoV. Examples include design 1 and 2 in Figure 1. *Near head space* closely ties to the robot's head around eyes. Examples include design 3 and 4 in Figure 1. *Transition space* includes designs that extend from the robot into its operating environment, such as design 5 in Figure 1. As the indicator moves closer to the task setting, we hypothesize that those designs will better help people to identify the performance effects of FoV. Finally, the designs grouped in *task space*, design 6 to 9 in Figure 1, are those not attached to the robot but rather placed in its working environment. Spectrum in Figure 2 offers a visual breakdown of our indicator designs, emphasizing the continuum from the robot space to the environment space.

## 4 DESIGNS: STRATEGIES TO INDICATE FOV

To indicate the robot's FoV, we designed the following nine strategies. The number prefixes are the same as in Figure 1.

**(1) Eye Socket**: We deepen the robot's eye sockets using an AR overlay at the existing eye sockets. It creates a shadowing effect at the robot's eyes. As the sockets deepen, they physically limit what angle the eyes can see, thus matching the cameras' FoV. Note this design is only possible with AR, as physical alteration is difficult.

**(2) Near-Eye Blocks**: We add blocks directly to the sides of the robot's eyes to functionally block those outside of the camera's horizontal FoV. This design is possible both physically and in AR.

**(3) Glasses**: Similar to the near-eye block but with aesthetic purposes and more familiar in daily life, we add a pair of glasses with solid temples to obscure those outside of the field of view. Note that this design can also be added both physically or with AR.

**(4) Near-eye Hands**: A straightforward approach that does not require any modifications or additions is for the robot to raise its hands directly to the sides of its eyes to act as visual indicators of the extent of its field of view. Practically, it can be used in a quick-start guide after the robot is shipped to the users without them wearing AR headsets. This design does not need AR or physical alteration.

**(5) Extended Blocks**: To more accurately show the range of the robot's FoV (e.g., which objects the robot cannot see), we connect

the blocks from the robot's head to the task environment, so that people know exactly how wide the robot can see. Note that this design can only be practically made possible with AR.

**(6) Blocks at Task**: A more task-centric design is to place the blocks to demonstrate the extent of the robot's field of view only in the robot's task environment, e.g., table. Unlike the last *Extended Blocks* design, this one is in the environment rather than connected to the robot. Note that this design can also only be placed with AR.

**(7) Projected Lines**: Rather than near-eye AR displays or physical alteration, this design uses an overhead projector to project lines onto the robot's operating environment to indicate the robot's FoV. This projected AR technology frees interactants from wearing head-mounted displays or holding phones or tablets, thus making it ergonomic and scalable to a crowd, beneficial in group settings.

**(8) Diminished Environment**: Diminished reality is a technique to remove real objects from a real scene as if the objects disappeared but actually the background of the objects were rendered. We thus propose to delete everything that the robot cannot see, leaving only what is within the robot's field of view. Note that this design requires AR and can be implemented by rendering part of the robot behind the out-of-view part of the table.

**(9) Dim Environment**: Rather than removing all the content that the robot cannot see, this design reduces the brightness of those content. Compared to diminished reality, we believe this design will help people maintain an awareness of the task environment. Similarly, this design also requires AR.

## 5 HYPOTHESES

As the **indicators are increasingly closer to the task space** (the right end of the spectrum in Figure 2), we believe they will bring task-related and subjective benefits. Thus, we develop the following four hypotheses.

**Hypothesis 1 (H1)**: Participants will develop a **more accurate mental model** of the robot's visual capability. This will be measured by the error rate, specifically, the percentage of requests made to the robot to hand over objects outside its FoV compared to those within. We expect a decrease in the error rate metric.

**Hypothesis 2 (H2)**: Indicators towards the environment will **improve task efficiency** more during human-robot collaborations because less time will be wasted to guess whether the robot can fulfill the requests or for the robots to ask clarification questions. Efficiency will be measured by the total time taken to complete collaborative tasks.

**Hypothesis 3 (H3)**: Participants will be **more confident** in the robot handing task objects. This will be measured by a seven-point Likert scale question.

**Hypothesis 4 (H4)**: Designs closer to the environment will require **less cognitive effort**. This will be measured by the well-established NASA Task Load Index [11, 21].

## 6 METHOD

To validate our hypotheses, we designed and plan to run a human-subjects study with a $1 \times 9$ within-subjects design. We will control the ordering and learning effects by a balanced Latin square design.

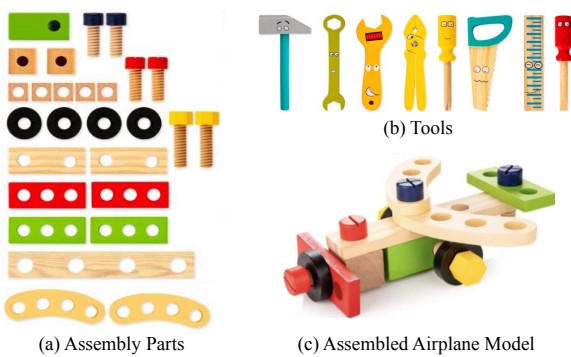

(b) Tools

(a) Assembly Parts

(c) Assembled Airplane Model

**Figure 3: The toolkit used in our collaborative task.**

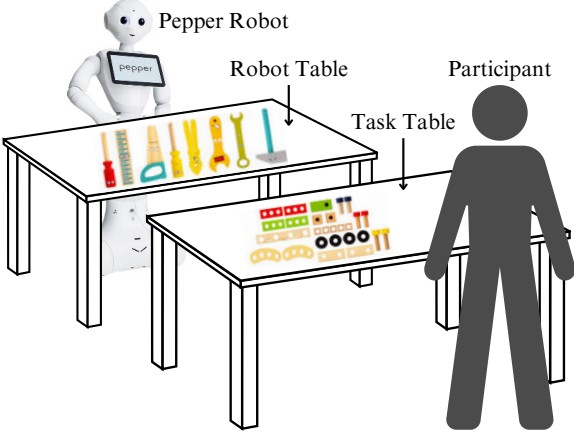

**Figure 4: Initial task setting. Assembly parts in Figure 3a are on the task table before the Pepper Robot. Tools in Figure 3b are on the task table in front of the participant.**

## 6.1 Apparatus and Materials

**Robot Platform**: We plan to use a Pepper robot [23] manufactured by Aldebaran. It is a two-armed, $1.2m$ ($3.9ft$) tall humanoid robot. Its narrow horizontal FoV is $54.4°$ [1].

**AR Display**: Participants will wear a Microsoft HoloLens 2 head-mounted display [20]. It has a $43° × 29°$ FoV.

**Toolkit Set**: A toolkit set [16] will be used for a toy airplane assembly task. It includes six types of tools (2 wrenches, 2 screwdrivers, 1 plier, 1 hammer, 1 saw, and 1 ruler) as shown in Figure 3b and five types of assembly parts (9 assembly pieces, 3 building blocks, 4 wheels, 6 bolts, and 5 nuts) as shown in Figure 3a.

**Room Setting**: As shown in Figure 4, we plan to set up the task with two tables: One table in front of the robot (robot table), and the other one for participants to complete the task (task table). All six tools (wrench, screwdriver, pliers, hammer, saw, and ruler) will be placed equally spaced on the robot table. The assembly parts (piece, block, wheel, bolt, nut) will be placed on the task table.

## 6.2 Task and Implementation

Participants will complete an airplane assembly task (shown in Figure 3) with the robot's help. As seen in Figure 3c, because the

screws can be manually tightened, participants will be asked to use the provided tools during assembly instead of hands. During the task, they can ask the robot to hand tools on the robot table.

We will develop these AR indicators in Unity except for design 4 (using near-eye hands) and design 7 (projected AR) in Figure 1. AR overlays will precisely match the physical dimensions and positions of the robot and the task-related objects. We will use the Vuforia Engine [26]'s tracking capability for registration. Particularly, we will register these designs to an image target pasted onto the robot, so that the indicators can be superimposed onto the physical robot and participants can see the indicators by wearing an AR headset.

For design 4, we plan to have the robot raise its hands before starting tasks. For design 7, we plan to place a projector on a turret over the robot's head and project lines onto the working environment during tasks, like the projected AR work by Han et al. [9].

## 6.3 Procedure

Upon arrival, participants will be provided with a consent form with the purpose of the study and the task. After agreeing to participate, they will complete a demographic survey. Then, they will be assigned to one of the ten Latin Square orderings over the ten experimental conditions (i.e., designs in Figure 1). We will conduct a training session to familiarize participants with HoloLens 2 and the task and to mitigate novelty effects, including watching how to wear HoloLens 2 and how to assemble the airplane. Before starting a task, experimenters will briefly reintroduce the task and ask final clarification questions. After experiencing a condition, they will complete a questionnaire about subjective metrics. At the end of the experiment when participants experience all conditions, they will be debriefed. Compensation will be determined in the future.

## 6.4 Data Collection and Measures

**Accuracy** will be calculated by error rate: the percentage of out-of-view requests among all requests. We will record videos and code the number of tool requests made by each participant. This includes the total number of requests as well as a separate count of requests for tools that are outside the robot's FoV. **Task Completion Time** will be coded from the videos from when the participants say they are ready and when participants say the assembly is complete. For the NASA Task Load Index [11, 21] measuring **cognitive effort**, we will use both the load survey and its weighting component to calculate a weighted average score. In this seven-point Likert scale to measure **confidence**, participants will be asked to indicate how confident their request would be fulfilled by the robot.

## 7 CONCLUSION

In this workshop paper, we proposed nine FoV indicator designs and detailed our initial experiment design. A task was designed to investigate the task-related performance and subjective experience with the designs to visually indicate a robot's physical vision capability. Our goal is to help align human expectations with a robot's actual capabilities for humans to form an accurate mental model of robots. Our immediate future work is to conduct the experiment, analyze the data, and report findings. We hope our results will provide insights into how robot designers can make better design decisions about indicating a robot's physical vision capability.

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
