# OpenReview forum: "To Understand Indicators of Robots' Vision Capabilities"
_humanrobotinteraction.org/HRI/2024/Workshop/VAM-HRI — VAM-HRI 2024 Oral_

### Official Review · Reviewer_9UpA · 2024-02-01
**Accept**

**Rating:** 7
**Confidence:** 5

**Review:**

This paper explores the design space of field of view (FoV) indicators in augmented reality (AR) to help humans accurately understand a robot's vision capabilities. It proposes nine innovative AR-based FoV indicators, aiming to align human mental models with the actual capabilities of robots during collaborative tasks. An experiment design is presented to evaluate these indicators in terms of accuracy, task efficiency, and subjective experience.

### Strengths:
- Innovative approach to addressing the misalignment of mental models between humans and robots.
- Comprehensive experimental design to assess the effectiveness of AR-based FoV indicators.
- Potential to significantly improve human-robot interaction by making robot capabilities more transparent.

### Weaknesses:
- The paper's results are based on planned experiments, and actual outcomes or data were not presented.
- The effectiveness of these indicators in diverse real-world scenarios remains to be validated.
- The reliance on AR technology might limit the applicability in environments where such technology is unavailable or impractical.

In summary, I think this paper is a great fit for VAM-HRI, and I recommend acceptance.

---

### Official Review · Reviewer_e1Yy · 2024-02-02
**Accept**

**Rating:** 6
**Confidence:** 5

**Review:**

This paper investigates the use of augmented reality (AR) technology to address the mental model misalignment between robots and humans, specifically focusing on the field of view (FoV). Motivated by the need to align human expectations with robots' actual vision capabilities, the paper introduces a design taxonomy categorizing indicators into four distinct areas: eye space, head space, transition space, and task space, forming a spectrum from robot to environment. Nine strategies namely, eye socket, near-eye blocks, glasses, near-eye hands, extended blocks, blocks at task, projected lines, diminished environment, and dim environment, are proposed to indicate robots' FoV. The paper outlines their definitions and discusses their applicability in both AR and real robot settings. A toy airplane assembly task is proposed to evaluate the strategies and further the practical application of the design.


Strengths:

-Clear problem statement- The paper addresses a specific problem related to human-robot interaction, focusing on the misalignment of mental models regarding robots' field of view (FoV).

-Interesting design strategy- The paper introduces a design spectrum and strategies, with nine proposed indicator designs based on their proximity to the robot's physical space and the task environment.

-Methodology and hypotheses: The experimental design and hypothesis are well described. The toy assembly task shows the potential of real-life applications and methods for improving the use of Social Robots using AR technology.



Weaknesses:

-The exploration of design spaces in AR could be further elaborated upon in the paper to provide a more comprehensive understanding of how these spaces are investigated to arrive at the proposed taxonomy and strategies.

-While the proposed design strategies show potential for indicating the field of view (FOV) in various scenarios, some strategies require further refinement to ensure their clarity and effectiveness. For instance, clarification is needed for the "diminished environment" strategy, specifying whether it refers to the real-world scene or the augmented scene. Addressing questions about the use of masking or post-processing camera input would enhance the practical implementation of these strategies.

-The absence of experiments makes it difficult to validate the hypotheses and strategies.

In summary, the paper is a good fit for VAM-HRI, and I recommend acceptance.

---

### Decision · Program_Chairs · 2024-02-06

Accept (Oral)